# Non-convolutional Graph Neural Networks

**Yuanqing Wang**
Center for Data Science
and Simons Center
for Computational Physical Chemistry
New York University
New York, N.Y. 10004
`wangyq@wangyq.net`

**Kyunghyun Cho**
Center for Data Science, New York University
and Prescient Design, Genetech
New York, N.Y. 10004
`kc119@nyu.edu`

## Abstract

Rethink convolution-based graph neural networks (GNN)—they characteristically suffer from limited expressiveness, over-smoothing, and over-squashing, and require specialized sparse kernels for efficient computation. Here, we design a simple graph learning module entirely free of convolution operators, coined *random walk with unifying memory* (RUM) neural network, where an RNN merges the topological and semantic graph features along the random walks terminating at each node. Relating the rich literature on RNN behavior and graph topology, we theoretically show and experimentally verify that RUM attenuates the aforementioned symptoms and is more expressive than the Weisfeiler-Lehman (WL) isomorphism test. On a variety of node- and graph-level classification and regression tasks, RUM not only achieves competitive performance, but is also robust, memory-efficient, scalable, and faster than the simplest convolutional GNNs.

## 1 Introduction: Convolutions in GNNs

Graph neural networks (GNNs) [1, 2, 3, 4, 5]—neural models operating on representations of nodes ($\mathcal{V}$) and edges ($\mathcal{E}$) in a *graph* (denoted by $\mathcal{G} = \{\mathcal{V}, \mathcal{E}\}, \mathcal{E} \subseteq \mathcal{V} \times \mathcal{V}$, with structure represented by the adjacency matrix $A_{ij} = \mathbb{1}[(v_i, v_j) \in \mathcal{E}]$)—have shown promises in a wide array of social and physical modeling applications. Most GNNs follow a *convolutional* scheme, where the $D$-dimensional node representations $\mathbf{X} \in \mathbb{R}^{|V| \times D}$ are aggregated based on the structure of local neighborhoods:

$$\mathbf{X}' = \hat{A}\mathbf{X}. \tag{1}$$

Here, $\hat{A}$ displays a unique duality—the input features doubles as a compute graph. The difference among *convolutional* GNN architectures, apart from the subsequent treatment of the resulting intermediate representation $\mathbf{X}'$, typically amounts to the choices of transformations ($\hat{A}$) of the original adjacency matrix ($A$)—the normalized Laplacian for graph convolutional networks (GCN) [1], a learned, sparse stochastic matrix for graph attention networks (GAT) [6], powers of the graph Laplacian for simplifying graph networks (SGN) [7], and the matrix exponential thereof for graph neural diffusion (GRAND) [8], to name a few. For all such transformations, it is easy to verify that permutation equivariance (Equation 9) is trivially satisfied, laying the foundations of data-efficient graph learning. At the same time, this class of methods share common pathologies as well:

**Limited expressiveness. (Figure 1)** Xu et al. [2] groundbreakingly elucidates that GNNs cannot exceed the expressiveness of Weisfeiler-Lehman (WL) isomorphism test [9]. Worse still, when the support of neighborhood multiset is uncountable, no GNN with a single aggregation function can be

38th Conference on Neural Information Processing Systems (NeurIPS 2024).

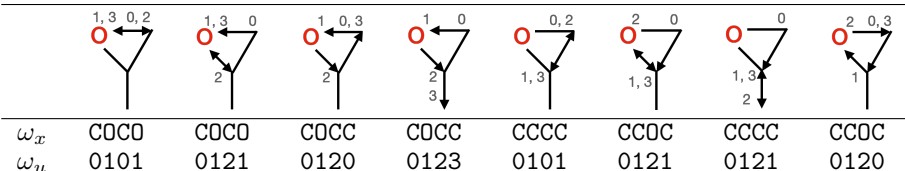

| | | | | | | | | |
|---|---|---|---|---|---|---|---|---|
| $\omega_x$ | COCO | COCO | COCC | COCC | CCCC | CCOC | CCCC | CCOC |
| $\omega_u$ | 0101 | 0121 | 0120 | 0123 | 0101 | 0121 | 0121 | 0120 |

Table 1: **Schematic illustration of RUM.** All 4-step unbiased random walks from the 2-degree carbon atom in the (hydrogen-omitted) chemical graph of *propylene oxide*, a key precursor for manufacturing polyurethane. The arrows indicate the direction of the walks and numbers the order in which each node is visited. The semantic ($\omega_x$) and topological ($\omega_u$) representations of each walk are shown.

as expressive as the WL test [10]. As such, crucial local properties of graphs meaningful in physical and social modeling, including cycle sizes (Example 8.1) and diameters (Example 8.2) [11], cannot be realized by convolution-based GNNs.

**Over-smoothing. (Figure 2)**    As one repeats the convolution (or Laplacian smoothing) operation (Equation 1), sandwiched by linear and nonlinear transformations, the inter-node dissimilarity, measured by Dirichlet energy,

$$\mathcal{E}(\mathbf{X}) = \frac{1}{N} \sum_{(u,v) \in \mathcal{E}_\mathcal{G}} ||\mathbf{X}_u - \mathbf{X_v}||^2, \tag{2}$$

will decrease exponentially as the number of message-passing steps $l$ increases [12, 13], $\mathcal{E}(\mathbf{X}^{(l)}) \leq C_1 \exp(-C_2 l)$ with some constants $C_{1,2}$, resulting in node representations only dependent upon the topology, but not the initial embedding.

**Over-squashing. (Figure 5)**    As the number of Laplacian smoothing grows, the receptive field of GNNs increases exponentially, while the dimension of node representation, and thereby the possible combinations of neighborhood environment, stays unchanged [14]. Quantitatively, Topping et al. [15] quantifies this insight using the inter-node Jacobian and relates it to the powers of the adjacency matrix through *sensitivity analysis*:

$$|\frac{\partial \mathbf{X}_v^{(l+1)}}{\partial \mathbf{X}_u^{(0)}}| \leq |\nabla \phi|^{(l+1)} (\hat{A}^{l+1})_{uv}, \tag{3}$$

where $\phi$ is the node-wise update function, whose Jacobian is typically diminishing. If this Equation 3 converges to zero, the node representation is agnostic to the changes happening $l$ edges away, making convolutional GNNs difficult to learn long-range dependencies.

**Main contributions: Non-convolutional GNNs as a joint remedy.**    In this paper, we propose a variant of GNN that does not engage the convolution operator (Equation 1) at all.[1] Specifically, we stochastically sample a random walk terminating at each node and use the *anonymous experiment* associated with the random walk to describe its topological environment. This is combined with the semantic representations along the walk and fed into a recurrent neural network layer [16] to form the node embedding, which we call the *unifying memory*. We theoretically (§ 4) and experimentally (§ 5) show that the resulting model, termed *random walk with unifying memory* (RUM) relieves the aforementioned symptoms and offers a compelling alternative to the popular convolution-based GNNs.

## 2   Related works: ways to walk on a graph

**Walk-based GNNs.**    RAW-GNN ([17], compared and outperformed in Table 9) also proposes walk-based representations for representing node neighborhoods, which resembles our model without the *anonymous experiment* ($\omega_u$ in Equation 5). CRaWl ([18, 19], outperformed in Table 3) also incorporates a similar structural encoding for random walk-generated subgraphs to feed into an

---

[1]Code at: `https://github.com/yuanqing-wang/rum/tree/main`

iterative 1-dimensional convolutional network. AWE ([20], Table 3) and Wang et al. [21], like ours, use anonymous experiments for graph-level unsupervised learning and temporal graph learning, respectively. More elaborately, [22] and AgentNet ([23], Table 3) use agent-based learning on random walks and paths on graphs.

**Random walk kernel GNNs.** In a closely related line of research, RWK ([24], Table 3) employs the reproducing kernel Hilbert space representations in a neural network for graph modeling and GSN ([25], Table 3) counts the explicitly enumerated subgraphs to represent graphs. The subgraph counting techniques intrinsically require prior knowledge about the input graph of a predefined set of node and edge sets. For these works, superior expressiveness has been routinely argued, though usually limited to a few special cases where the WL test fails whereas they do not, and often within the *unlabelled graphs* only.

More importantly, focusing mostly on expressiveness, no aforementioned **walk-based** or **random walk kernel**-based GNNs address the issue of over-smoothing and over-squashing in GNNs. Some of these works are also *de facto* convolutional, as the random walks are only incorporated as features in the message-passing scheme. Interestingly, most of these works are either not applicable to, or have not been tested on, node-level tasks. In the experimental § 5, we show that RUM not only outperforms these baselines on most graph-level tasks (Table 3) but also competes with a wide array of state-of-the-art convolutional GNNs on node-level tests (Table 2). Moreover, random walk kernel-based architectures, which explicitly enumerate random walk-generated subgraphs are typically much slower than WL-equivalent convolutional GNNs, whereas RUM is faster than even the simplest variation of convolutional GNN (Figure 4).

**Graph transformers.** Graph transformers [26, 27]—neural models that perform attention among all pairs of nodes and encode graph structure via positional encoding—are well-known solutions that are not locally convolutional. Their inductive biases determine that over-smoothing and over-squashing among local neighborhoods are, like RUM, also not prominent.

Because of its all-to-all nature, the runtime complexity of graph transformers, just like that of almost all transformers, contains a quadratic term, w.r.t. the size of the system (number of nodes, edges, or subgraphs). This makes it prohibitively expensive and memory intensive on large social graphs (such as that used in Table 8 with millions of nodes). On smaller social graphs, we show in Tables 9, 4 that graph transformers are experimentally outperformed by RUM.

**State-of-the-art methods to alleviate oversmoothing.** *Stochastic regularization.* DropEdge ([28], Figure 2) regularizes the smoothing of the node representations by randomly disconnecting edges. Its associated Dirichlet energy indeed decreases slower, though eventually still diminishes as the number of layers increases. *Graph rewiring.* [29, 30] and GPR-GNN ([31], Appendix Table 9) rewire the graph using personalized page rank algorithm [32] and generalized page rank on graphs, respectively. Similar to JKNet [33], they mitigate over-smoothing by allowing direct message passing between faraway nodes. *Constant-energy methods.* Zhao and Akoglu [34], Rusch et al. [35] constrain the pair-wise distance or Dirichlet energy among graphs to be constant. Nevertheless, the non-decreasing energy does not necessarily translate to better performance, as they sometimes come with the sacrifice of expressiveness, as argued in Rusch et al. [13]. *Residual GNNs.* Residual connection [36] can naturally be added to the GNN architecture, such as GCNII ([37], Table 2), to restraint activation to be similar to the input to allow deep networks. They however can make the model less robust to perturbations in the input. In sum, these works have similar, if not compromised expressiveness compared to a barebone GCN.

## 3 Architecture: combining topologic and semantic trajectories of walks

**Random walks on graphs.** An unbiased random walk $w$ on a graph $\mathcal{G}$ is a sequence of nodes $w = (v_0, v_1, \ldots)$ with landing probability:

$$P(v_j | (v_0, \ldots, v_{i-1})) = \mathbb{1}[(v_i, v_j) \in \mathcal{E}_{\mathcal{G}}]/D(v_i), \qquad (4)$$

where $D(v_i) = \sum A_{ij}$ is the degree of the node $v_i$. Walks *originating* from or *terminating* at any given node $v$ can thus be easily generated using this Markov chain. We record the trajectory of embeddings associated with the walk as $\omega_x(w) = (\mathbf{X}_i) = (\mathbf{X}_0, \mathbf{X}_1, \ldots, \mathbf{X}_l)$. In this paper, we only

consider finite-long $l$-step random walk $|w| = l \in \mathbb{Z}^+$. In our implementation, the random walks are sampled *ad hoc* during each training and inference step directly on GPU using Deep Graph Library [38] (see Appendix § A). Moreover, the *walk* considered here is not necessarily a *path*, as repeated traversal of the same node $v_i = v_j, i \neq j$ is not only permitted, but also crucial to effective topological representation, as discussed below.

**Anonymous experiment.** We use a function describing the topological environment of a walk, termed *anonymous experiment* [39], $\omega_u(w) : \mathbb{R}^l \to \mathbb{R}^l$ that records **the first unique occurrence of a node in a walk** (Appendix Algorithm C). To put it plainly, we label a node as the number of *unique* nodes insofar traversed in a walk if the node has not been traversed, and reuse the label otherwise. Practically, this can be implemented using any tensor-accelerating framework *in one line* (w is the node sequence of a walk) and trivially parallelized [2]: `(1*(w[..., :,None]==w[..., None,:])).argmax(-1)`

**Unifying memory: combining semantic and topological representations.** Given any walk $w$, we now have two sequences $\omega_x(w)$ and $\omega_u(w)$ describing the *semantic* and *topological* (as we shall show in the following sections) features of the walk. We project such sequential representations onto a latent dimension to combine them (illustrated in Table 1):

$$h(w) = f(\phi_x(\omega_x(w)), \phi_u(\omega_u(w))), \tag{5}$$

where $\phi_x : \mathbb{R}^{l \times D} \to \mathbb{R}^{D_x}$ maps the sequence of semantic embeddings generated by a $l$-step walk to a fixed $D_x$-dimensional latent space, $\phi_u : \mathbb{R}^l \to \mathbb{R}^{D_u}$ maps the indicies sequence to another latent space $D_u$, and $f : \mathbb{R}^{D_x} \oplus \mathbb{R}^{D_u} \to \mathbb{R}^D$ combines them. We call Equation 5 the *unifying memory* of a random walk. Subsequently, the node representations can also be formed as the average representations of $l$-step ($l$ being a hyperparameter) walks *terminating* (for the sake of gradient aggregation) at that node:

$$\psi(v) = \sum_{\{w\}, |w|=l, w_l=v} p(w)h(w), \tag{6}$$

which can be stochastically sampled with unbiased Monte Carlo gradient and used in downstream tasks as such node classification and regression. We note that this is the only time we perform SUM or MEAN operations. Unlike other GNNs incorporating random walk-generated features (which are sometimes still convolutional and iterative), we do not iteratively pool representations within local neighborhoods. The likelihood of the data written as:

$$P(y|\mathcal{G}, \mathbf{X}) = \sum_{\{w\}, |w|=l, w_l=v} p(w)p(y|\mathcal{G}, \mathbf{X}, w) \tag{7}$$

The node representation can be summed

$$\Psi(\mathcal{G}) = \sum_{v \in \mathcal{V} \subseteq \mathcal{G}} \psi(v) \tag{8}$$

to form global representations for graph classification and regression. We call $\psi$ in Equation 6 and $\Psi$ in Equation 8 the node and graph output representations of RUM.

**Layer choices.** Obvious choices to model $f$ include a feed-forward neural network after concatenation, and $\phi_x, \phi_u$ recurrent neural networks (RNNs). This implies that, different from most convolutional GNNs, parameter sharing is natural and the number of parameters is going to stay constant as the model incorporates a larger neighborhood. Compared to dot product-based, transformer-like modules [40], RNNs not only have linear complexity (see detailed discussions below) w.r.t. the sequence length but also naturally encodes the inductive bias that nodes closer to the origin have stronger impact on the representation. The gated recurrent unit (GRU)[16] variant is used everywhere in this paper. Additional regularizations are described in Appendix § B.1.

---

[2]Note that this particular implementation introduces an intermediate $\mathcal{O}(l^2)$ complexity term, though it is empirically faster than the linear-complexity naive implementation, since only integer indices are involved and thus the footprint is negligible.

**Runtime complexity.**     To generate random walks for one node has the runtime complexity of $\mathcal{O}(1)$, and for a graph $\mathcal{O}(|\mathcal{V}|)$, where $|\mathcal{V}|$ is the number of nodes in a graph $\mathcal{G} = \{\mathcal{V}, \mathcal{E}\}$. To calculate the *anonymous experiment*, as shown in Appendix Algorithm C, has $\mathcal{O}(1)$ complexity (also see Footnote 2). If we use linear-complexity models, such as RNNs, to model $\phi_x, \phi_u$, the overall complexity is $\mathcal{O}(|\mathcal{V}|lkD)$ where $l$ is the length of the random walk, $k$ the samples used to estimate Equation 6, and $D$ the latent size of the model (assumed uniform). Note that different from convolutional GNNs, RUM does not depend on the number of edges $|\mathcal{E}|$ (which is usually much larger than $|\mathcal{V}|$) for runtime complexity, and is therefore agnostic to the *sparsity* of the graph. See Figure 4 for an empirical verification of the time complexity. In Appendix Table 8, we show, on a large graph, the overhead introduced by generating random walks and computing anonymous experiments accounts for roughly $1/1500$ of the memory footprint and $1/8$ of the wall time.

**Mini-batches.**    RUM is naturally compatible with mini-batching. For convolutional GNNs, large graphs that do not fit into the GPU memory have traditionally been a challenge, as all neighbors are required to be present and boundary conditions are hard to define [41]. RUM, on the other hand, can be inherently applied on subsets of nodes of a large graph without any alteration in the algorithm—the random walks can be generated on a per-node basis, and the FOR loop in Algorithm C can be executed sequentially, in parallel, or on subsets. Empirically, in Appendix Table 8, RUM can be naturally scaled to huge graphs.

# 4   Theory: RUM as a joint remedy.

We have insofar designed a new graph learning framework—convolution-free graph neural networks (GNNs) that cleanly represent the semantic ($\omega_x$) and topological ($\omega_u$) features of graph-structured data before unifying them. First, we state that RUM is permutation equivariant,

*Remark* 1 (Permutation equivariance). For any permutation matrix $P$, we have

$$P\mathbf{X}_v(\mathcal{G}) = \mathbf{X}_v(P(\mathcal{G})),\tag{9}$$

which sets the foundation for the data-efficient modeling of graphs. Next, we theoretically demonstrate that this formulation jointly remedies the common pathologies of the convolution-based GNNs by showing that: (a) the *topological* representation $\omega_u$ is more expressive than convolutional-GNNs in distinguishing non-isomorphic graphs; (b) the *semantic* representation $\omega_x$ no longer suffers from over-smoothing and over-squashing.

## 4.1   RUM is more expressive than convolutional GNNs.

For the sake of theoretical arguments in this section, we assume that in Equation 5:

**Assumption 2.**  $\phi_x, \phi_u, f$ are universal and injective.

**Assumption 3.**  Graph $\mathcal{G}$ discussed in this section is always connected, unweighted, and undirected.

Assumption 2 is easy to satisfy for feed-forward neural networks [42] and RNNs [43]. Note that RUM can be easily extended to weighted graphs by sampling a biased random walk with edge weights $w_{ij}$ and keeping rest of the algorithm the same: $P(v_j|(v_0, ..., v_{i-1})) \propto I[(v_i, v_j) \in E_G] * w_{ij}/D(v_i)$. Composing injective functions, we remark that $h(w)$ is also injective w.r.t. $\omega_x(w)$ and $\omega_u(w)$; despite of Assumption 3, our analysis can be extended to disjointed graphs by restricting the analysis to the connected regions in a graph. Under such assumptions, we show, in **Remark** 8 (deferred to the Appendix), that $\psi$ is **injective**, meaning that nodes with different random walks will have different distributions of representations $\psi(v_1) \neq \psi(v_2)$. We also refer the readers to the **Theorem 1** in Micali and Zhu [39] for a discussion on the representation power of *anonymous experiments* on *unlabelled* graphs. Combining with the semantic representations and promoting the argument from a node level to a graph level, we arrive at:

**Theorem 4** (RUM can distinguish non-isomorphic graphs). *Up to the Reconstruction Conjecture [44], RUM with sufficiently long $l$-step random walks can distinguish non-isomorphic graphs satisfying Assumption 3.*

The main idea of the proof of Theorem 4 (in Appendix § D.2) involves explicitly enumerating all possible non-isomorphic structures for graphs with 3 nodes and showing, by induction, that if

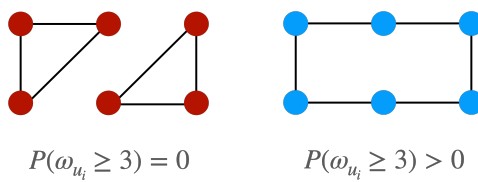

$$P(\omega_{u_i} \geq 3) = 0 \qquad P(\omega_{u_i} \geq 3) > 0$$

Figure 1: RUM can (in closed form), whereas the Weisfeiler-Lehman (WL) isomorphism test and WL-equivalent GNNs *cannot*, distinguish these two graphs—**an illustration of Example 8.1**.

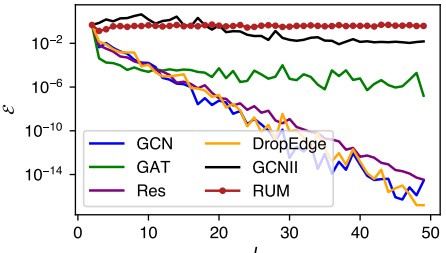

Figure 2: **RUM alleviates over-smoothing**. Dirichlet energy ($\mathcal{E}$) on Cora [47] graph plotted against $L$, the number of steps or layers.

the theorem stands for graph of $N - 1$ size it also holds for $N$-sized graphs. We also show in Appendix § D.1 that a number of key graph properties such as cycle size (Example 8.1) and radius (Example 8.2) that convolutional GNNs struggle [45, 46] to learn can be analytically expressed using $\omega_u$. As these are solely functions of $\omega_x$, they can be approximated arbitrarily well by universal approximators. These examples are special cases of the finding that RUM is stricly more expressive than Weisfeiler-Lehman isomorphism test [9]:

**Corollary 4.1** (RUM is more expressive than WL-test). *Up to the Reconstruction Conjecture, two graphs with $G_1, G_2$ labeled as non-isomorphic by the $k$-dimensional Weisfeiler-Lehman ($k$-WL) isomorphism test, is the necessary, but not sufficient condition that the representations resulting from RUM with walk length $k$ are also different.*

$$\Psi(\mathcal{G}_1) \neq \Psi(\mathcal{G}_2) \tag{10}$$

Thus, due to Xu et al. [2], RUM is also more expressive than convolutional GNNs in distinguishing non-isomorphic graphs. This also confirms the intuition that RUM with longer walks are more expressive. (See Figure 3 on an empirical evaluation.) The proof of this corollary is straightforward to sketch—even if we only employ the embedding trajectory $\omega_x$, it would have the effect of performing the function $\phi_x$ in Equation 5 on each traversal of the WL expanded trees.

### 4.2 RUM alleviates over-smoothing and over-squashing

Over-smoothing refers to the phenomenon where the node dissimilarity (e.g., measured by Dirichlet energy in Equation 2) decreases exponentially and approaches zero with the repeated rounds of message passing. Cai and Wang [12] relates Dirichlet energy directly with the convolutional operator:

**Lemma 3.1 from Cai and Wang [12].**

$$\mathcal{E}((1 - \tilde{\Delta})\mathbf{X}) \leq (1 - \lambda)^2 \mathcal{E}(\mathbf{X}) \tag{11}$$

*where $\lambda$ is the smallest non-zero eigenvalue of $\tilde{\Delta}$, the normalized Laplacian of a graph.*

Free of convolution operators, it seems only natural that RUM does not suffer from this symptom (Figure 2). We now formalize this intuition by first restricting ourselves to a class of *non-contractive* mappings for $f$ in Definition 5.

**Definition 5.** A map $f$ is non-contractive on region $\Omega$ if $\exists \alpha \in [1, +\infty)$ such that $|f(x) - f(y)| \geq \alpha ||x - y||, \forall x, y \in \Omega$.

A line of fruitful research has been focusing on designing non-contractive RNNs [48, 49], and to be non-contractive is intimately linked with desirable properties such as preserving the long-range information content and non-vanishing gradients. From this definition, it is easy to show that, for each sampled random walk in Equation 5, the Dirichlet energy is greater than its input. One only needs to verify that the integration in Equation 6 does not change this to arrive at:

**Lemma 6** (RUM alleviates over-smoothing.). *If $\phi_x, f$ are non-contractive w.r.t. all elements in the sequence, the expected Dirichlet energy of the corresponding RUM node representation in Equation 6 is greater than its initial value*

$$\mathrm{E}(\mathcal{E}(\psi(\mathbf{X}))) \geq \mathcal{E}(\mathbf{X}). \tag{12}$$

This implies that the expectation of Dirichlet energy does not diminish even when $l \to +\infty$, as it is bounded by the Dirichlet energy of the initial node representation, which is consistent with the trend shown in Figure 2, although the GRU is used out-of-box without constraining it to be explicitly non-contractive.

**RUM alleviates over-squashing**  is deferred to Appenxix § B.2, where we verify that the inter-node Jacobian $|\partial \mathbf{X}_v^{(l+1)}/\partial \mathbf{X}_u^{(0)}|$ decays slower as the distance between $u, v$ grows vis-à-vis the convolutional counterparts. Briefly, although RUM does not address the information bottleneck with exponentially growing receptive field (the $1/(\hat{A}^{l+1})_{uv}$ term in Equation 16), it nevertheless can have a non-vanishing (nor exploding) gradient from the aggregation function ($|\nabla \phi_x|$).

# 5    Experiments

On a wide array of real-world node- and graph-level tasks, we benchmark the performance of RUM to show its utility in social and physical modeling. Next, to thoroughly examine the performance of RUM, we challenge it with carefully designed illustrative experiments. Specifically, we ask the following questions in this section, with **Q1**, **Q2**, and **Q3** already theoretically answered in § 4: **Q1:** Is RUM more expressive than convolutional GNNs? **Q2:** Does RUM alleviate over-smoothing? **Q3:** Does RUM alleviate over-squashing? **Q4:** Is RUM slower with convolutional GNNs? **Q5:** Is RUM robust? **Q6:** How does RUM scale up to huge graphs? **Q7:** What components of RUM are contributing most to the performance of RUM?

**Real-world benchmark performance.**  For node classification, we benchmark our model on the popular Planetoid citation datasets [47], as well as the coauthor [50] and co-purchase [51] datasets common in social modeling. Additionally, we hypothesize that RUM, without the smoothing operator, will perform competitively on heterophilic datasets [52]—we test this hypothesis. For graph classification, we benchmark on the popular TU dataset [53]. We also test the graph regression performance on molecular datasets in MoleculeNet [54] and Open Graph Benchmark [55]. In sum, RUM almost always outperforms, is within the standard deviation of, the state-of-the-art architectures, as shown in Tables 2, 3, 4, 5, as well as in Tables 6, 7, 8, 9 moved to the Appendix due to space constraint.

**On sparsity: the subpar performance on the Computer dataset.**  The most noticeable exception to the good performance of RUM is that on the Computer co-purchase [51] dataset, where RUM is outperformed even by GCN and GAT. This dataset is very dense with an average node degree ($|\mathcal{E}|/|\mathcal{V}|$) of $18.36$, the highest among all datasets used in this paper. As the variance of the node embedding (Equation 6) scales with the average degree, we hypothesize that dense graphs with very high average node degrees would have high-variance representations from RUM.

On the other hand, RUM outperforms *all* models surveyed in two large-scale benchmark studies on molecular learning, GAUCHE [56] and MoleculeNet [54]. The atoms in the molecules always have a degree of $2 \sim 4$ with intricate subgraph structures like small rings. This suggests the utility of *unifying memory* in chemical graphs and furthermore chemical and physical modeling.

**Graph isomorphism testing (Q1).**  Having illustrated in § 4 that RUM can distinguish non-isomorphic graphs, we experimentally test this insight on the popular Circular Skip Link dataset [60, 61]. Containing 4-regular graph with edges connected to form a cycle and containing skip-links between nodes, this dataset is artificially synthesized in Murphy et al. [60] to create an especially challenging task for GNNs. As shown in Appendix Table 7, all convolutional GNNs fail to perform better than a constant baseline (there are 10 classes uniformly distributed). 3WLGNN [62], a higher-order GNN of at least $\mathcal{O}(2)$ complexity that operates on explicitly enumerated triplets of graphs, can distinguish these highly similar 4-regular graphs by comparing subgraphs. RUM, with $\mathcal{O}(|\mathcal{N}|)$ linear complexity, achieves similarly high accuracy. One can think of RUM as a stochastic approximation

|  | Cora | CiteSeer | PubMed | Coathor CS | Computer | Photo |
|---|---|---|---|---|---|---|
| GCN[1] | 81.5 | 70.3 | 79.0 | $91.1_{\pm0.5}$ | $82.6_{\pm2.4}$ | $91.2_{\pm1.2}$ |
| GAT[6] | $83.0_{\pm0.7}$ | $72.5_{\pm0.7}$ | $79.0_{\pm0.3}$ | $90.5_{\pm0.6}$ | $78.0_{\pm19.0}$ | $85.7_{\pm20.3}$ |
| GraphSAGE[4] | $77.4_{\pm1.0}$ | $67.0_{\pm1.0}$ | $76.6_{\pm0.8}$ | $85.0_{\pm1.1}$ |  | $90.4_{\pm1.3}$ |
| MoNet[57] | $81.7_{\pm0.5}$ | $70.0_{\pm0.6}$ | $78.8_{\pm0.4}$ | $90.8_{\pm0.6}$ | $83.5_{\pm2.2}$ | $91.2_{\pm2.3}$ |
| GCNII[37] | $85.5_{\pm0.5}$ | $73.4_{\pm0.6}$ | $80.3_{\pm0.4}$ |  |  |  |
| PairNorm[34] | 81.1 | 70.6 | 78.2 |  |  |  |
| GraphCON[35] | $84.2_{\pm1.3}$ | $74.2_{\pm1.7}$ | $79.4_{\pm1.3}$ |  |  |  |
| RUM | $84.1_{\pm0.9}$ | $75.5_{\pm0.5}$ | $82.2_{\pm0.2}$ | $93.2_{\pm0.0}$ | $77.8_{\pm2.3}$ | $92.7_{\pm0.1}$ |

Table 2: **Node classification** test set accuracy ↑ and standard deviation.

|  | IMDB-B | MUTAG | PROTEINS | PTC | NCI1 |
|---|---|---|---|---|---|
| RWK[24] |  | $79.2_{\pm2.1}$ | $59.6_{\pm0.1}$ | $55.9_{\pm0.3}$ |  |
| GK[58] |  | $81.4_{\pm1.7}$ | $71.4_{\pm0.3}$ | $55.7_{\pm0.5}$ | $62.5_{\pm0.3}$ |
| WLK[59] | $73.8_{\pm3.9}$ | $90.4_{\pm5.7}$ | $75.0_{\pm3.1}$ | $59.9_{\pm4.3}$ | $86.0_{\pm1.8}$ |
| AWE[20] | $74.5_{\pm5.9}$ | $87.8_{\pm9.8}$ |  |  |  |
| GIN[2] | $75.1_{\pm5.1}$ | $90.0_{\pm8.8}$ | $76.2_{\pm2.6}$ | $66.6_{\pm6.9}$ | $82.7_{\pm1.6}$ |
| GSN[25] | $77.8_{\pm3.3}$ | $92.2_{\pm7.5}$ | $76.6_{\pm5.0}$ | $68.2_{\pm7.2}$ | $83.5_{\pm2.0}$ |
| CRaWl[19] | $73.4_{\pm2.1}$ | $90.4_{\pm7.1}$ | $76.2_{\pm3.7}$ | $68.0_{\pm6.5}$ |  |
| AgentNet[23] | $75.2_{\pm4.6}$ | $93.6_{\pm8.6}$ | $76.7_{\pm3.2}$ | $67.4_{\pm5.9}$ |  |
| RUM | $81.1_{\pm4.5}$ | $91.0_{\pm7.1}$ | $77.3_{\pm3.8}$ | $69.8_{\pm6.3}$ | $81.7_{\pm1.4}$ |

Table 3: **Binary graph classification** test set accuracy ↑.

of the higher-order GNN, with all of its explicitly enumerated subgraphs being identified by RUM with a probability that decreases with the complexity of the subgraph.

**Effects of walk lengths and number of samples on performance (Q2, Q3).** Having studied the relationship between inference speed and the walk lengths and number of samples, we furthermore study its impact on performance. Using Cora [47] citation graph and vary the walk lengths and number of samples from 1 to 9, where the performance of RUM improves as more samples are taken and longer walks are employed, though more than 4 samples and walks longer than $L > 4$ yield qualitatively satisfactory results; this empirical finding has guided our hyperparameter design. In Figure 2, we also compare the Dirichlet energy of RUM-generated layer representations with not only baselines GCN [1] and GAT [6], but also strategies to alleviate over-smoothing discussed in § 2, namely residual connection and stochastic regularization [37, 28, 63], and show that when $L$ gets large, only RUM can maintain Dirichlet energy. Traditionally, since Kipf and Welling [1] (see its Figure 5 compared to Figure 3), the best performance on Cora graph was found with 2 or 3 message-passing rounds, since mostly local interactions are dominating the classification, and more rounds of message-passing almost always lead to worse performance. As theoretically demonstrated in § 4.2, RUM is not as affected by these symptoms. Thus, RUM is especially appropriate for modeling long-range interactions in graphs without sacrificing local representation power.

|  | Texas | Wisc. | Cornell |
|---|---|---|---|
| GCN[1] | $55.1_{\pm4.2}$ | $51.8_{\pm3.3}$ | $60.5_{\pm4.8}$ |
| GAT[6] | $52.2_{\pm6.6}$ | $51.8_{\pm3.1}$ | $60.5_{\pm5.3}$ |
| GCNII[37] | $77.6_{\pm3.8}$ | $80.4_{\pm3.4}$ | $77.9_{\pm3.8}$ |
| Geom-GCN[52] | $66.8_{\pm2.7}$ | $64.5_{\pm3.7}$ | $60.5_{\pm3.7}$ |
| PairNorm[34] | $60.3_{\pm4.3}$ | $48.4_{\pm6.1}$ | $58.9_{\pm3.2}$ |
| GPS[26] | $75.4_{\pm1.5}$ | $78.0_{\pm2.9}$ | $65.4_{\pm5.7}$ |
| Graphomer [27] | $76.8_{\pm1.8}$ | $77.7_{\pm2.0}$ | $68.4_{\pm1.7}$ |
| RUM | $80.0_{\pm7.0}$ | $85.8_{\pm4.1}$ | $71.1_{\pm5.6}$ |

Table 4: **Node classification** test set accuracy ↑ and standard deviation on heterophilic [52] datasets.

|  | ESOL | FreeSolv | Lipophilicity |
|---|---|---|---|
| GAUCHE[56] | $0.67_{\pm0.01}$ | $0.96_{\pm0.01}$ | $0.73_{\pm0.02}$ |
| MoleculeNet [54] | 0.58 | 1.15 | 0.80 |
| RUM | $0.62_{\pm0.06}$ | $0.96_{\pm0.24}$ | $0.66_{\pm0.01}$ |

Table 5: **Graph regression** RMSE ↓ compared with the *best* model studied in two large-scale benchmark studies on OGB [55] and MoleculeNet [54] datasets.

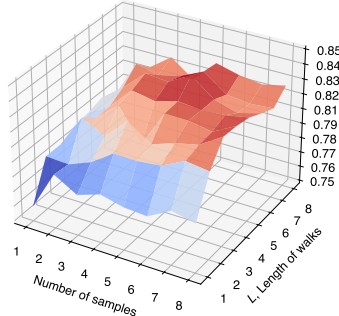

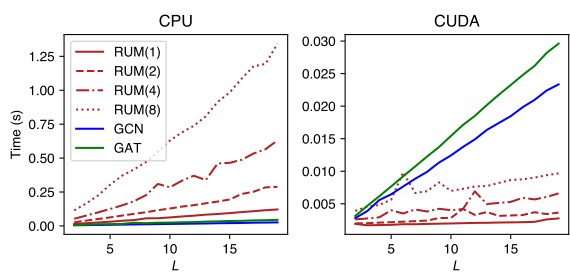

Figure 3: **Impact of number of samples and walk length.** Test classification accuracy of Cora [47] with varying numbers of samples and walk length.

Figure 4: **RUM is faster than convolutional GNNs on GPU.** Inference time over the Cora [47] graph on CPU and CUDA devices, respectively, plotted against $L$, the number of message-passing steps or equivalently the length of random walks. Numbers in the bracket indicate the number of sampled random walks drawn.

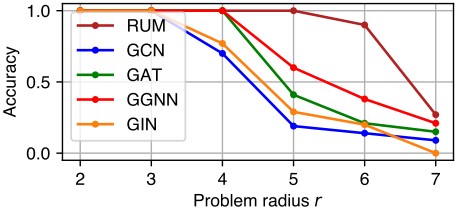

Figure 5: **Long-range neighborhood matching** training accuracy ↑ [14] with 32 unit models.

Figure 6: **Robustness analysis**. Accuracy ↑ on Cora [47] dataset with % fictitious edges added to the graph.

**Long-range neighborhood matching (Q3).** To verify that RUM indeed alleviates over-squashing (§ B.2), in Figure 5, we adopt the tree neighborhood matching synthetic task in Alon and Yahav [14] where binary tree graphs are proposed with the label *and the attributes* of the root matching a faraway leave. The full discussion is moved to the Appendix § B.3.

**Speed (Q4).** Though both have linear runtime complexity (See § 3), intuitively, it might seem that RUM would be slower than convolution-based GCN due to the requirement of multiple random walk samples. This is indeed true for CPU implementations shown in Figure 4 left. When equipped with GPUs (specs in Appendix § A), however, RUM is significantly faster than even the simplest convolutional GNN—GCN [1]. It is worth mentioning that the GCN and GAT [6] results were harvested using the heavily optimized Deep Graph Library [38] sparse implementation whereas RUM is implemented naïvely in PyTorch [64], though the popular GRU component [16] have already undergone CUDA-specific optimization.

**Robustness to attacks (Q5).** With the stochasticity afforded by the random walks, it is natural to suspect RUM to be robust. We adopt the robustness test from Feng et al. [65] and attack by randomly adding fake edges to the Cora [47] graph and record the test set performance in Figure 6. Indeed, RUM is much more robust than traditional convolutional GNNs including GCN [1] and GAT [6] and is even slightly more robust than the convolutional GNN specially designed for robustness [65], with the performance only decreased less than 10% with 10% fake edges added.

**Scaling to huge graphs (Q6) and Ablation study (Q7)** are deferred to Appendix § B.5.

# 6 Conclusions

We design an innovative GNN that uses an RNN to unify the semantic and topological representations along stochastically sampled random walks, termed *random walk with unifying memory* (RUM) neural networks. Free of the convolutional operators, our methodology does not suffer from symptoms characteristic of Laplacian smoothing, including limited expressiveness, over-smoothing, and over-squashing. Most notably, our method is more expressive than the Weisfeiler-Lehman isomorphism test and can distinguish all non-isomorphic graphs up to the *reconstruction conjecture*. Thus, it is more expressive than all of convolutional GNNs equivalent to the WL test, as we demonstrate theoretically in § 4. RUM is significantly faster on GPUs than even the simplest convolutional GNNs (§ 5) and shows superior performance across a plethora of node- and graph-level benchmarks.

**Limitations.** *Very dense graphs.* As evidenced by the underwhelming performance of the Computer [51] dataset and discussed in § 5, RUM might suffer from high variance with very dense graphs (average degree over 15). *Tottering.* In our implementation, we have not ruled out the 2-cycles from the random walks, as that would require specialized implementation for walk generation. This, however, would reduce the average information content in a fixed-length random walk (known as tottering [66]). We plan to characterize the effect of excluding these walks. *Biased walk.* Here, we have only considered unbiased random walk, whereas biased random walk might display more intriguing properties as they tend to explore faraway neighborhoods more effectively [17]. *Directed graphs.* Since we have only developed RUM for undirected graph (random walk up to a random length is not guaranteed to exist for directed graphs), we would have to symmetrize the graph to perform on directed graphs (such as the heterophilic datasets [52]); this create additional information loss and complexity.

**Future directions.** *Theoretical.* We plan to expand our theoretical framework to account for the change in layer width and depth to derive analytical estimates for realizing key graph properties. *Applications.* Random walks are intrinsically applicable to uncertainty-aware learning. We plan to incorporate the uncertainty naturally afforded by the model to design active learning models. On the other hand, the physics-based graph modeling field is also heavily dominated by convolutional GNNs. Inspired by the superior performance of RUM on chemical datasets, we plan to apply our method in drug discovery settings [67, 68, 69, 70, 71] and furthermore on the equivariant modeling of $n$-body physical systems [72].

**Impact statement.** We here present a powerful, robust, and efficient learning algorithm on graphs. Used appropriately, this algorithm might advance the modeling of social [73] and physical [74] systems, which can oftentimes modeled as graphs. As with all graph machine learning methods, negative implications may be possible if used in the design of explosives, toxins, chemical weapons, and overly addictive recreational narcotics.

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

# A Experimental details

**Code availability.** All architectures, as well as scripts to execute the experiment, are distributed open-source under MIT license at `https://anonymous.4open.science/r/rum-834D/`. Core dependencies of our package include PyTorch [64] and Deep Graph Library [38].

**Hyperparameters.** All models are optimized using Adam [75] optimizer and SiLU [76] activation functions. 4 random walk samples are drawn everywhere unless specified. Other hyperparameters—learning rate ($10^{-5} \sim 10^{-2}$), hidden dimension ($32 \sim 64$), L2 regularization strength ($10^{-8} \sim 10^{-2}$), walk length ($3 \sim 16$), temperature for $\mathcal{L}_{\texttt{consistency}}$ ($0 \sim 1$), coefficient for $\mathcal{L}_{\texttt{consistency}}$ ($0 \sim 1$), coefficient for $\mathcal{L}_{\texttt{self}}$, and dropout probability—are tuned using the Ray platform [77] with the default Ax [78] search algorithm with 1000 trails or 24 hours tuning budget on a Nvidia A100® GPU.

# B Additional technical details

## B.1 Self-supervised regularization.

The stochasticity encoded in our model naturally affords it with some level of regularization. Apart from using the consistency loss ($\mathcal{L}_{\texttt{consistency}}$) used in Feng et al. [65] for classifications, we further regularize the model by using the RNNs in $\phi_x$ to predict the semantic representation of the next node on the walk given $\omega_u$ and jointly maximize this likelihood:

$$\hat{\omega}_{x_{i+1}} = g(\{\omega_{x_1}, \omega_{x_2}, \ldots, \omega_{x_i}\}, \omega_u | \theta); \tag{13}$$

$$\mathcal{L}_{\texttt{self}}(\theta) = -\log P(\hat{\omega}_{x_{i+1}} | \theta), \tag{14}$$

where $g(\cdot | \theta)$ is modeled as the sequence output of the RNN $\phi_x$ in Equation 5. The total loss is modeled as a combination of:

$$\mathcal{L}(\theta) = -\log P(y | \mathcal{G}, \mathbf{X}, \theta) + \mathcal{L}_{\texttt{self}} + \mathcal{L}_{\texttt{consistency}} \tag{15}$$

## B.2 RUM attenuates over-squashing

Similarly, we can show that if the composing neural networks defy the vanishing gradient problem (w.r.t. the input) [79], the *sensitivity analysis* in Equation 3 [15] has a lower bound for RUM.

**Lemma 7** (RUM attenuates over-squashing). *If $\phi_x, f$ have lower-bounded derivatives, the inter-node Jacobian for nodes $u, v$ separated by a shortest path of length $l$, RUM with walk length $l$ also has a lower bound:*

$$\left| \frac{\partial \mathbf{X}_v^{(l)}}{\partial \mathbf{X}_u^{(0)}} \right| \geq |\nabla \phi_x||\nabla f|(\hat{A}^l)_{uv}, \tag{16}$$

*where $\hat{A}_{ij} = A_{ij} / \sum_j A_{ij}$ is the degree-normalized adjacency matrix.*

Like the upper bound in Equation 3, this lower bound is also controlled by the power of the (normalzied) adjacency matrix, albeit the absence of self-loop will result in a slightly looser bottleneck. The term $(\hat{A}^l)_{uv}$ corresponds to the probability of the shortest path among all possible walks as a product of inverse node degrees (see Equation 4). There is no denying that the lower bound is still controlled by the power of the adjacency matrix, which corresponds to the exponentially growing receptive fields. One can also argue that, without prior knowledge, the contribution of the sensitivity analysis by the power of the graph adjacency matrix can never be alleviated, since there are always roughly $1/(\hat{A}^{l+1})_{uv}$ (assuming uniform node degree) structurally equivalent nodes. Nevertheless, since $\phi_x$ is not necessarily an iterative function, we alleviate the over-squashing problem by eliminating the power of the update function gradient term.

Now, we plug in the layer choices of $\phi_x$—a GRU [16] unit. Its success, just like that of long short-term memory (LSTM) [80], can be attributed to the near-linear functional form of the long-range gradient. The term $|\nabla \phi_x|$ is controlled by a sequence of sigmoidal update gates, which can be optimized to approach 1 (fully open). If we ignore the gradient contribution of $\mathbf{X}_u^0$ to these gates, the non-linear activation function has only been applied exactly *once* on $\mathbf{X}_u^0$; therefore, the gradient $|\partial \mathbf{X}_v^{(l+1)} / \partial \mathbf{X}_u^{(0)}|$ is neither rapidly vanishing nor exploding.

| $\omega_u = \mathbf{0}$ | $\omega_x = \mathbf{0}$ | $\mathcal{L}_{\texttt{self}} = 0$ | $\mathcal{L}_{\texttt{consistency}} = 0$ |
|---|---|---|---|
| $82.2 \pm 1.0$ | $35.0 \pm 1.0$ | $78.4 \pm 0.1$ | $80.3 \pm 1.1$ |

Table 6: **Ablation study.** Cora [47] test set accuracy $\uparrow$ with in the architecture deleted.

### B.3 Long-range neighborhood matching (Q3).

Once identifying the target leaf, this task seems trivial; nonetheless, this piece of information needs to be passed through layers of aggregation and non-linear update and is usually lost in the convolution. Since, on this binary tree, the receptive field grows exponentially, Alon and Yahav [14] argues that there is a theoretical lower boundary for the layer width $D$ for the convolutional GNN to be able to encode all possible combinations of leaves, which is $2^{32D}$ for single-precision floating point (`float32`). This corresponds to the structural $\hat{A}^l$ term in Equation 3 and Equation 16. Evidently, when $D = 32$, as is the adopted experimental setting in Figure 5, this limit is far from being hit. So we hypothesize that the reason why convolutional GNNs cannot overfit the training set is because of the limitation of the functional forms, which are remedied by RUM, which shows $100\%$ accuracy up to tree depth or problem radius $r = 5$, and a relatively moderate decrease afterward. Note that when the problem radius exceeds $r = 7$, RUM's performance is not significantly different from the convolutional counterparts.

### B.4 Scalaing to large graphs (Q6).

In Appendix Table 8, we apply RUM on an ultra-large graph `OGB-PRODUCTS`, that cannot fit easily on a single GPU, and compare RUM with architectures specifically designed for large graphs [41].

### B.5 Ablation study (Q7).

In Table 6, we conduct a brief ablation study where we delete, one by one, the components introduced in § 3. $\omega_u = \mathbf{0}$ and $\omega_x = \mathbf{0}$ refer to the deletion of the topological and semantic representations of walks, respectively. Neglecting topological information results in a moderate decrease in performance, whereas neglecting semantic representation is more detrimental. The $\omega_u = \mathbf{0}$ also resembles Jin et al. [17] albeit with different walk-wise aggregation. This offers a qualitative comparison between our work and Jin et al. [17] as no source code was released for this package so no rigorous comparison was possible. We also see that the regularization methods are helpful to the performance, with self-supervision being more crucial. We attribute this effect to firstly the relative simplicity of the Cora classification task, and secondly the flexibility of the (overparametrized) RNNs.

## C   Additional results

---
**Algorithm 1** anonymous experiment

---
   **Input:** $w = (v_0, v_1, \ldots, v_l)$
   $C \leftarrow 0; \Omega \leftarrow \text{Dict}(\{\})$
   **for** $v_i$ in $w$ **do**
      **If** $v_i$ in $\Omega$: $u_i \leftarrow \Omega[v_i]$; **Else** $\Omega[v_i] \leftarrow C; C \leftarrow C + 1$
   **end for**
   **Return:** $\omega_u = (u_i) = (u_0, u_1, \ldots, u_l)$

---

*Remark* 8 (Inequality in distribution). For two nodes $v_1, v_2$ with distribution of random walks terminating at $v_1, v_2$ not equal in distribution $p(\omega_u(w_1)) \neq p(\omega_u(w_2))$ or $p(\omega_x(w_1)) \neq p(\omega_x(w_2))$, the node representations in Equation 6 are also different $\psi(v_1) \neq \psi(v_2)$.

One way to construct $h$ function in Equation 5 is to have $h(w)$ positive only where $p(\omega_u(w_1)) > p(\omega_u(w_2))$; the same thing can be argued for $\omega_x$. In other words, one only needs to prove two walks terminating at two nodes $p(\omega_u(w_1)) \neq p(\omega_u(w_2))$ or $p(\omega_x(w_1)) \neq p(\omega_x(w_2))$ are not *equal in distribution* to verify that RUM can distinguish two nodes. Conversely, we can also show that $p(\omega_u(w_1)) = p(\omega_u(w_2))$ implies that $v_1, v_2$ are isomorphic without labels— we refer the readers to the **Theorem 1** in Micali and Zhu [39] for a discussion on reconstructing unlabelled *graphs* using anonymous experiments.

|  | Complexity | CSL accuracy |
|---|---|---|
| GCN[1] | $\mathcal{O}(N)$ | $10.0 \pm 0.0$ |
| GAT[6] | $\mathcal{O}(N)$ | $10.0 \pm 0.0$ |
| GIN[2] | $\mathcal{O}(N)$ | $10.0 \pm 0.0$ |
| GraphSAGE[4] | $\mathcal{O}(N)$ | $10.0 \pm 0.0$ |
| 3WLGNN[62] | $\mathcal{O}(N^2)$ | $95.7 \pm 14.8$ |
| RUM | $\mathcal{O}(N)$ | $93.2 \pm 0.8$ |

Table 7: Graph classification accuracy ↑ on CSL [60] synthetic dataset for graph isomorphism test.

|  | Accuracy | Memory (MB) | Throughput(iter/s) |
|---|---|---|---|
| GraphSAGE [4] | $80.61 \pm 0.16$ | 415.94 | 37.69 |
| ClusterGCN [81] | $78.62 \pm 0.61$ | 10.62 | 156.01 |
| GraphSAINT [82] | $75.36 \pm 0.34$ | 10.95 | 143.51 |
| FastGCN [83] | $73.46 \pm 0.20$ | 11.54 | 93.05 |
| LADIES [84] | $73.51 \pm 0.56$ | 20.33 | 93.47 |
| SGC [7] | $67.48 \pm 0.11$ | 0.01 | 267.31 |
| SIGN [85] | $76.85 \pm 0.56$ | 16.21 | 208.52 |
| SAGN [86] | $81.21 \pm 0.07$ | 71.81 | 80.04 |
| RUM | $76.1 \pm 0.50$ | 47.64 | 119.93 |
| w/o walks |  | 47.56 | 139.45 |
| only walks |  | 0.03 | 950.66 |

Table 8: **Node classification accuracy and efficiency** on `OGB-PRODUCTS` [55]

|  | # params | Cora | Photo |
|---|---|---|---|
| GCN [1] | 48K | $87.14 \pm 1.01$ | $88.26 \pm 0.83$ |
| GAT [6] | 49K | $88.03 \pm 0.79$ | $90.04 \pm 0.68$ |
| GCNII [37] | 49K | $88.46 \pm 0.82$ | $89.94 \pm 0.31$ |
| RAW-GNN |  | $87.85 \pm 1.52$ |  |
| LanczosNet [87] | 50K | $87.77 \pm 1.45$ | $93.21 \pm 0.85$ |
| GPR-GNN [31] | 48K | $88.57 \pm 0.69$ | $93.85 \pm 0.28$ |
| PP-GNN [88] |  | $89.52 \pm 0.85$ | $92.89 \pm 0.37$ |
| Transformer | 37K | $71.83 \pm 1.68$ | $90.05 \pm 1.50$ |
| Graphomer [27] | 139K | $67.71 \pm 0.78$ | $95.20 \pm 4.12$ |
| Specformer [89] | 32K | $88.57 \pm 1.01$ | $95.48 \pm 0.32$ |
| RUM | 23K +20K initial proj. | $89.01 \pm 1.40$ | $95.35 \pm 0.26$ |

Table 9: **Node classification** test accuracy ↑ and standard deviation with 60:20:20 random split.

# D  Missing mathematical arguments.

## D.1  Examples of Theorem 4

**Example 8.1** (Cycle detection.). *A $k$-cycle $\mathcal{C}_k$ is a subgraph of $\mathcal{G}$ consisting of $k$ nodes, each with degree two. The existence of $k$-cycle can be determined by:*

$$\mathbb{1}(\mathcal{C}_k \subseteq \mathcal{G}) = \mathbb{1}[P(\omega_{x_{j+1}} = \omega_{x_0}, \omega_{x_i} \neq \omega_{x_j}, \forall i < j) > 0] \tag{17}$$

**Example 8.2** (Diameter.). *The diameter, $\delta_{\mathcal{G}}$ of graph $\mathcal{G}$ which equals the length of the longest shortes path in $\mathcal{G}$, can be expressed as*

$$\delta_{\mathcal{G}} = \underset{l = |\omega_x|, \omega_{x_i} \neq \omega_{x_j}, \forall i \neq j}{\mathrm{argmax}} |\omega_x| \tag{18}$$

## D.2  Proof of Theorem 4

*Proof.* First, we enumerate all possible *unlabelled* graphs with three nodes satisfying Assumption 3— one with two edges, one with three edges. (Note that there is only one non-isomorphic graph with

two nodes.) Now we consider random walks of length $l = 3$, where

$$P(\omega_{u_3} = \omega_{u_0}) > 0 \tag{19}$$

only stands for the graph with three edges, but not with two edges, just like Example 8.1.

Furthermore, we can also distinguish between the 2-degree node and the 1-degree node in the graph with 2 edges and 3 nodes simply by verifying that

$$P(\omega_{u_1} \neq \omega_{u_3}) > 0 \tag{20}$$

only stands when $v_2$ is the 2-degree node.

Moving on to the labeled 3-node graph case, we can reduce the problem to investigate whether RUM can distinguish 3-node graphs that are isomorphic when unlabeled, but non-isomorphic when labeled. For the three-edged graph, $\omega_x$ uniformly samples the labels of three nodes. For the two-edged graph, suppose the node labels are $A, B, C$, and we start from the node with 2 degrees $B$ (with nodes bearing $A$ and $C$ labels locally, structurally isomorphic),

$$P(B|\omega_x(w_t)) = \begin{cases} 1, t = 2n, n \in \mathbb{N}, \\ 0, t = 2n + 1, n \in \mathbb{N} \end{cases} \tag{21}$$

$$P(A|\omega_x(w_t)) = P(C|\omega_x(w_t)) = \begin{cases} 0, t = 2n, n \in \mathbb{N}, \\ 1/2, t = 2n + 1, n \in \mathbb{N} \end{cases} \tag{22}$$

If graphs have the same $\omega_x$, they have the same $B$ and the same or swapped $A, C$. As such, we have proven Theorem 4 for graphs with 3 nodes.

Now we prove that Theorem 4 stand for graphs of $N$ nodes, they also stand for graphs of $N + 1$ nodes, the Reconstruction Conjecture [44]

Suppose we have two non-isomorphic graphs with $N + 1$ nodes $\mathcal{G}_1$ and $\mathcal{G}_2$ with the same RUM embedding $\Psi_1 = \Psi_2$. We enumerate all $N + 1$ subgraphs with each node deleted for each of these two graphs. By the Reconstruction Conjecture, at least one pair of subgraphs are non-isomorphic. For this pair, suppose the deleted vertex is $v$ (ruling out the trivial case where the label or connectivity of $v$ is different for these two graphs), and two remaining subgraphs $\mathcal{G}_1^{\setminus v}, \mathcal{G}_2^{\setminus v}$; since $\phi_x, \phi_u, f$ are injective, $\Psi_1 = \Psi_2$ implies $\omega_u, \omega_x$ are *equal in distribution* for $\mathcal{G}_1, \mathcal{G}_2$. As such, the walk distribution

$$P(\omega_x(w), \omega_u(w)|w_0 = v, w_i \neq v, i > 0) \tag{23}$$
$$= P(\omega_x(w_0), \omega_u(w_0)|w_0 = v)P(\omega_x(w_{1...}), \omega_u(w_{1...})|\omega_x(w_{1...}), \omega_u(w_{1...}), w_i \neq v, i > 0) \tag{24}$$

are also *equal in distribution* for $\mathcal{G}_1, \mathcal{G}_2$.

If there is a link between $v$ and the nodes in $\mathcal{G}_1, \mathcal{G}_2$, we assign a new label to contain both the old label and the connection. As such, if the second term is not equal in distribution for $\mathcal{G}_1, \mathcal{G}_2$, we would have

$$\Psi(\mathcal{G}_1^{\setminus v}) \neq \Psi \mathcal{G}_2^{\setminus v}), \tag{25}$$

which is in conflict with the assumption that Theorem 4 stands for graphs with $N$ nodes. □

### D.3 Proof of Lemma 6

*Proof.* By the definition of Dirichlet energy (Equation 2) and non-contractive mappings (Definition 5),

$$\mathcal{E}(\psi(\mathbf{X})) = \frac{1}{N} \sum_{u,v \in \mathcal{E}_\mathcal{G}} ||\psi(\omega_x(\mathbf{X}_u)) - \psi(\omega_x(\mathbf{X}_v))||^2 \tag{26}$$

$$= \frac{1}{N} \sum_{u,v \in \mathcal{E}_\mathcal{G}} ||f(\phi_x(\omega_x(\mathbf{X}_u))) - f(\phi_x(\omega_x(\mathbf{X}_v)))||^2 \tag{27}$$

$$\geq \frac{1}{N} \sum_{u,v \in \mathcal{E}_\mathcal{G}} ||\phi_x(\omega_x(\mathbf{X}_u)) - \phi_x(\omega_x(\mathbf{X}_v))||^2 \tag{28}$$

$$= \frac{1}{N} \sum_{u,v \in \mathcal{E}_\mathcal{G}} ||\phi_x( \sum_{\{w\},|w|=l,w_l=u} p(w)\mathbf{X}_u, u \in w) - \phi_x( \sum_{\{w\},|w|=l,w_l=v} p(w)\mathbf{X}_v, v \in w)||^2 \tag{29}$$

$$\geq \frac{1}{N} \sum_{u,v \in \mathcal{E}_\mathcal{G}} ||( \sum_{\{w\},|w|=l,w_l=u} p(w)\mathbf{X}_u, u \in w) - ( \sum_{\{w\},|w|=l,w_l=v} p(w)\mathbf{X}_v, v \in w)||^2 \tag{30}$$

$$\geq \mathcal{E}(\mathbf{X}) \tag{31}$$

The last inequality is due to the fact that $\phi_x$ is non-contractive w.r.t. all, and therefore, the last element of the sequence, which are always $\mathbf{X}_u$ and $\mathbf{X}_v$. $\qquad\square$

