# OpenReview forum: "Non-convolutional graph neural networks."
_NeurIPS.cc/2024/Conference — NeurIPS 2024 spotlight_

### Official Review · Reviewer_jBUd · 2024-06-26

**Soundness:** 3
**Presentation:** 2
**Contribution:** 3
**Rating:** 6
**Confidence:** 2

**Summary:**

This paper proposes a random walk-based graph neural network, where an RNN is used to combine the topological and semantic graph features along the walks. The proposed RUM model is free of convolution operators and does not suffer from the limited expressiveness, over-smoothing, and over-squashing, which are commonly faced issues in convolution-based GNNs.

**Strengths:**

This paper proposes a non-convolutional GNN, named RUM. The benefits of this model over convolutional GNNs are elaborated both theoretically and empirically.

**Weaknesses:**

Empirically, the gains with RUM are marginal and in some cases, RUM is worse than the baselines.

The organization and the presentation of the paper can be further improved.
1. There are too many hyper-references. Most of the hyper-references in Section 1 and Section 2 are unnecessary, especially when reading the paper for the first time.
2. There are too many supplementary demonstrations (...) Some additional information could be helpful, but too many will hinder the reading. It would be better to reorganize the sentences to make the presentation smoother.
3. There are some grammar mistakes, like in line 244. Proofreading is needed.

**Questions:**

1. In Lemma 6, the author shows that RUM increases Dirichlet energy. Why would this be helpful intuitively? As for homophily graphs, GNNs should decrease the Dirichlet energy. This phenomenon is also analyzed in [1,2]. It seems RUM is not just alleviating over-smoothing, but is anti-smoothing.
2. In line 243, the author claims that RUM almost always outperforms SOTA architectures, this is not true according to the results.

[1] Interpreting and unifying graph neural networks with an optimization framework

[2] A Unified View on Graph Neural Networks as Graph Signal Denoising

**Limitations:**

The authors adequately discussed the limitations.

---

> ### Author Rebuttal · Authors · 2024-07-31
>
> Thank you, Reviewer `jBUd`, for your thoughtful and constructive review. We also thank you for highlighting the theoretical and empirical benefits. We hope that the following points address your concerns.
>
> ## Comparison with state-of-the-art models.
> > Empirically, the gains with RUM are marginal and in some cases, RUM is worse than the baselines.
>
> >In line 243, the author claims that RUM almost always outperforms SOTA architectures, this is not true according to the results.
>
> Among the 8 tables totaling 21 benchmark experiments and more than 100 comparisons, the only cases where RUM is not state-of-the-art by a confidence interval are:
> - In **Table 2**, Computers dataset and **Table 3**, NCI1 dataset. As discussed in sections **On sparsity** and **Limitations**, RUM faces challenges on very dense graphs.
> - In **Appendix Table 8**, GNNs specifically designed for large graphs outperform RUM, which works out-of-the-box for both large and small datasets with high efficiency and a modest memory footprint, and is naturally compatible with sub-graph mini-batches.
> - In **Table 4**, RUM is singularly outperformed by GCNII on one of the three tasks.
>
> As such, we feel like the wording of "almost always" is not inaccurate. Nevertheless, we still plan to tune down the statement and put the *competitive* performance of RUM in the context of the following facts.
> - **Efficiency.** RUM is faster than even the simplest convolutional GNNs on GPUs.
> - **Robustness** naturally afforded by the stochasticity. (Figure 6)
>
> Note that the competitive performance is further demonstrated in the new results provided in the rebuttal period (see the PDF).
>
> ## Clarification on over-smoothing
> >In Lemma 6, the author shows that RUM increases Dirichlet energy. Why would this be helpful intuitively?
>
> In Lemma 6, by $E(e(\Phi(X)) \geq e(X)$ under some conditions, we state that RUM is *possible* to *maintain non-decreasing, non-diminishing* Dirichlet energy *when non-contractive* functions (RNNs in this paper) $\phi_x, f$ are prescribed. It does not state that RUM always increases the Dirichlet energy, nor does it state that RUM is anti-smoothing.
>
> In other words, the smoothing behavior of RUM is controlled by the backbone RNN prescribed. All in all, if we want a smoothing RUM, we can couple it with a contractive RNN (if we use contractive activation functions among sequences); an expansive RNN would indeed lead to an anti-smoothing, gradient-exploding RUM (according to the definitions in \[1\]). For vanilla GRUs with sigmoidal updates that are approximately non-contractive (nor expansive), RUM would have roughly constant Dirichlet energy, which is in agreement with our empirical finding in Figure 2.
>
> In comparison, also shown in Figure 2 is that all traditional GNNs are destined to have diminishing Dirichlet energy as a result of the iterative convolutions (Laplacian smoothing), so it is impossible to design deep convolutional GNNs without resulting in very similar neighborhoods.
>
> To summarize, in Lemma 6 we want to show that RUM is *possible to be not over-smoothing* rather than always *anti-smoothing.* We do realize that the wording in this section confuses our purpose and will revise it. We thank you again for catching this issue.
>
> ## Presentation clarity.
> > The organization and the presentation of the paper can be further improved.
>
> > There are too many hyper-references.
>
> > There are too many supplementary demonstrations
>
> We apologize for the somewhat convoluting reading experience of this version of the manuscript as it has been significantly condensed and distilled to fit in the page limit. As such, we are disheartened to see that many substantial theoretical and empirical results, have to be moved to the Appendix, including 4 tables, 4 theoretical arguments, and the entire Appendix Section B, which contains important discussions on over-squashing, scaling, and ablation, hence the abuse of hyper-references.
>
> Meanwhile, we had hoped that Section 1 would serve not only as an introduction, but also as a problem statement and a directory. Similarly, Section 2 would relink the display elements scattered around the manuscript to form a thorough comparison between RUM and traditional, convolution-based RNNs. This attempt has clearly failed to provide the desired result and we plan to straighten the confusing sentences and cut down many of the hyper-references and supplementary demonstrations. We thank the reviewer again for bringing our attention to the presentation clarity.
>
> We hope that the discussion above has addressed your concern and that you will consider raising the score to see our manuscript in the next round with significantly improved representation and a clearer discussion on the Dirichlet energy behaviors.
>
> References:
> \[1\] Yoshua Bengio, Patrice Simard, and Paolo Frasconi. Learning long-term dependencies with gradient descent is difficult. IEEE transactions on neural networks, 5(2):157–166, 1994.

---

> ### Author Response · Authors · 2024-08-08
> **Clarification on the over-smoothing behavior**
>
> Dear Reviewer `jBUd`,
>
> Please let us know if our rebuttal has addressed your concerns, especially about the clarification on the over-smoothing behavior.
>
> Thank you!

---

> > ### Comment · Reviewer_jBUd · 2024-08-10
> > **Official Comment by Reviewer jBUd**
> >
> > Thank you for the detailed responses. Your clarifications relieve most of my concerns. I have increased the score.

---

> > > ### Author Response · Authors · 2024-08-10
> > > **Thank you!**
> > >
> > > Thank you, Reviewer `jBUd`, for your comment and for raising the score.
> > >
> > > Please don't hesitate to let us know if you have any further questions!

---

### Official Review · Reviewer_xhjS · 2024-07-07

**Soundness:** 3
**Presentation:** 3
**Contribution:** 3
**Rating:** 6
**Confidence:** 3

**Summary:**

The paper introduces Random Walk with Unifying Memory (RUM), a non-convolutional graph neural network (GNN) that addresses limitations like limited expressiveness, over-smoothing, and over-squashing typically associated with convolution-based GNNs. RUM leverages random walks and recurrent neural networks (RNNs) to merge topological and semantic graph features, theoretically showing and experimentally verifying its enhanced expressiveness over traditional methods like the Weisfeiler-Lehman (WL) isomorphism test. It demonstrates competitive performance on node- and graph-level tasks while offering advantages in efficiency and scalability.

**Strengths:**

1. RUM introduces a new paradigm in GNNs by eliminating convolutions and leveraging random walks and RNNs for enhanced graph representation.

2. The presentation of the paper is generally clear and coherent. The authors effectively contextualize their work relative to existing literature, highlighting the novelty of RUM and its contributions to the field of graph representation learning.

3. The theoretical advancements (e.g., expressiveness proofs) and empirical results clearly demonstrate the advantages of RUM over traditional convolutional GNNs. The findings are valuable for advancing graph representation learning techniques and are likely to influence future research directions.

**Weaknesses:**

The paper lacks detailed exploration on how varying random walk lengths impact RUM's performance across diverse tasks and whether optimal lengths are identified for different types of graphs or datasets. Additionally, while the paper highlights RUM's superiority over expressive GNNs in specific scenarios, it overlooks potential limitations when compared to these models, such as not thoroughly investigating upper bounds on the WL test, which could gauge RUM's discriminative power against more expressive GNNs.

**Questions:**

1. Can you elaborate on how the choice of random walk length $𝑙$ impacts the performance of RUM across different tasks? Are there optimal lengths identified for specific types of graphs or datasets?

2. In practical applications, how does RUM handle graph datasets with varying degrees of sparsity? Are there specific strategies employed to ensure robust performance across different graph structures?

3. While the paper demonstrates that RUM outperforms expressive GNNs in certain scenarios, it lacks exploration of scenarios where RUM might face challenges or limitations compared to expressive GNNs. For instance, it does not thoroughly investigate potential upper bounds on the WL test, which is a standard measure of graph isomorphism and could provide insights into the discriminative power of RUM compared to more expressive GNNs.

**Limitations:**

Yes, The authors have addressed several limitations in the Limitations section.

---

> ### Author Rebuttal · Authors · 2024-07-31
>
> Many thanks, Reviewer `xhjS`, for your insightful and constructive review. We also thank you for emphasizing the clarity of our theoretical framework and its potential impact on future research. We address your questions point-by-point as follows.
>
>
> ## The impact of random walk lengths.
> > The paper lacks detailed exploration on how varying random walk lengths impact RUM's performance across diverse tasks and whether optimal lengths are identified for different types of graphs or datasets.
>
> > Can you elaborate on how the choice of random walk length  impacts the performance of RUM across different tasks? Are there optimal lengths identified for specific types of graphs or datasets?
>
> When choosing $L$, as noted in the **Appendix: Experimental Details**, we tune it as a hyperparameter with a range of 3 to 16 for each dataset. The only exception is in the synthetic example of Figure 5, where $L$ is set according to the problem radius $r + 1$. Empirically, citation datasets usually display small-world cluster behavior, hence a smaller optimal $L$, whereas in molecular graphs, optimal $L$ is usually larger since there exist long-term dependencies and rings.
>
> Figure 3 can be seen as a hyperparameter grid search on a citation dataset (Cora). We add a hyperparameter sweep for a molecular dataset (ESOL) in the PDF.
>
> Apart from the impact of the hyperparameter $L$, the length of the random walk and the size of the receptive field, have been discussed in the following sections:
> - Corollary 4.1: The impact of $L$ on expressiveness. RUM with $L = N + 1$ is sufficient to distinguish non-isomorphic graphs up to size $N$.
> - Figure 2: The impact of $L$ on Dirichlet energy.
> - Figure 3: The impact of $L$ on experimental performance.
> - Figure 4: The impact of $L$ on efficiency.
>
> We also plan to add the following discussion in **Section 3: Architecture** to provide a recipe for narrowing the search space for picking $L$ for each problem _a priori_.
>
> ---
> The hyperparameter $L$, the length of the random walk, is an important factor that dictates the size of the receptive field and controls the speed-performance and bias-variance tradeoffs. A higher $L$ affords us a richer, more expressive, more expensive, and potentially higher-variance representation of a larger node neighborhood. We provide the following heuristics for picking the optimal $L$:
> - If the problem radius is known *a priori*, $L$ should match it. In the synthetic setting in Figure 5, $L$ is set to match the problem radius $r+1$.
> - If there is rough knowledge about the approximate problem radius (how long-term the interactions are), one can also use such knowledge to confine the search space to the region around it. For example, it is well-known that citation graphs such as that in Figure 3 manifest small-world behavior, whereas physical and chemical graphs sometimes contain long-range dependencies.
> - If no such information is available, one can resort to the traditional hyperparameter tuning (one grid search example is replayed in Figure 3). If there is previous experience in tuning the round of message passing for convolutional GNNs, the tuning experiments can start in the vicinity of the optimal solution. Nonetheless, we also note that larger $L$ does not necessarily cause deteriorated performance as evidenced by Figures 2 and 3, and it is possible that the optima for RUM might be larger than that of convolutional GNNs.
> ---
>
>
> ## The impact of sparsity.
> > In practical applications, how does RUM handle graph datasets with varying degrees of sparsity?
>
> > While the paper demonstrates that RUM outperforms expressive GNNs in certain scenarios, it lacks exploration of scenarios where RUM might face challenges or limitations compared to expressive GNNs.
>
> In the **Limitations** section as well as the **On sparsity** paragraph in **Section 5: Experiments**, we discuss that *very* dense graphs (such as the Computer dataset with an above 18 average node degree), are where RUM would face challenges, as the variance of the random walk, and thereby the resulting representation, is intrinsically higher. On molecules with an average 2~4 node degree, RUM is more performant even than the best models surveyed in large benchmark studies.
>
> > Are there specific strategies employed to ensure robust performance across different graph structures?
>
> As shown in Appendix Section B1, two forms of regularization are employed to ensure consistency:
> - Self-supervise: An additional loss is added for the model to predict the semantic embedding of the next node in the walk given its _anonymous experiment_
> - Consistency regularization: Variance among predictions given by different random walk samples is penalized.
>
>
> ## Expressiveness comparison with the WL test
> > For instance, it does not thoroughly investigate potential upper bounds on the WL test, which is a standard measure of graph isomorphism and could provide insights into the discriminative power of RUM compared to more expressive GNNs.
>
> We would also be grateful if you could provide more details on the meaning of "upper bounds of the WL test." Does it mean the dimension $k$ in the $k$-WL test \[1\]?
>
> If that is the case, in **Theorem 4**, we find that RUM can already distinguish *any* non-isomorphic graphs (or subgraphs) up to the Reconstruction Conjecture. As such, we can rewrite **Corollary 4.1** in a more quantitative manner, that RUM with $k$ -length walks are at least as powerful as $k$-WL:
>
> **Corollary 4.1 (RUM is more expressive than k-WL-test.)** Up to the _Reconstruction Conjecture_, two graphs with $G_1 , G_2$ labeled as non-isomorphic by the $k$-dimensional Weisfeiler-Lehman ($k$-WL) isomorphism test, is the necessary, but not sufficient condition that the representations resulting from RUM with walk length $k$ are also different.
>
> References:
> \[1\] Morris et al. 2018 Weisfeiler and Leman Go Neural: Higher-order Graph Neural Networks

---

> > ### Author Response · Authors · 2024-08-07
> > **Clarification on the concept of "upper bounds of the WL test"**
> >
> > Hi Reviewer `xhjS`,
> >
> > Thanks again for your feedback! We had a question on the meaning of "upper bounds of the WL test", does it mean the dimension of the $k$-WL test? If that's the case, in the rebuttal we had reworked our **Corollary 4.1** to incorporate a comparison with the $k$-WL test.
> >
> > Please let us know if we have interpreted your comments correctly. We will provide more thorough comparison and discussion to further address your comments if that is not the case.
> >
> > Thank you!

---

> > > ### Author Response · Authors · 2024-08-10
> > > **Correction: to remove the length condition in the new Corollary 4.1**
> > >
> > > After careful checking, we find that the condition of walk length $k$ has to be removed in the new Corollary 4.1 reworked during the rebuttal. (See this [post](https://openreview.net/forum?id=JDAQwysFOc&noteId=GFBlZllWUU))
> > >
> > > We would be grateful if you could cast more light on the concept of "upper bounds of the WL test" and whether a comparison with the $k$-WL test would be sufficient.

---

> > > > ### Comment · Reviewer_xhjS · 2024-08-10
> > > > **Satisfied with the answers**
> > > >
> > > > Thank you for your update and for revisiting Corollary 4.1. I appreciate the authors' detailed response. I am satisfied with the answers and will maintain my original score.

---

> > > > > ### Author Response · Authors · 2024-08-11
> > > > > **Thank you!**
> > > > >
> > > > > Thanks again, Reviewer `xhjS`!
> > > > >
> > > > > Please let us know if you have any further questions, or if you think of more ways to characterize the WL upper bound.

---

### Official Review · Reviewer_wXB7 · 2024-07-12

**Soundness:** 3
**Presentation:** 3
**Contribution:** 3
**Rating:** 7
**Confidence:** 4

**Summary:**

The paper introduced a new graph neural network that is not based on convolution but on the random walk. Based on the results of the extensive experiment, the proposed architecture is effective for heterophilic graphs and long-range graphs.

**Strengths:**

1. The experiment results include different tasks on different graphs, also, it is faster than baselines.
2. Theoretical analysis is conducted to show that the proposed architecture is more expressive.
3. Theoretical analysis shows that the proposed architecture is effective in overcoming over-squashing and over-smoothing issues.

**Weaknesses:**

Currently, the proposed architecture only works on unweighted and undirected graphs.

**Questions:**

Can the proposed method work well on directed and weighted graphs?

---

> ### Author Rebuttal · Authors · 2024-07-31
>
> Thank you, Reviewer `wXB7`, for your constructive and encouraging review and for highlighting that RUM is both theoretically innovative and experimentally efficient and performant. We address your question on the directed and weighted graphs as follows, which we plan to incorporate in the manuscript more clearly.
>
> > Currently, the proposed architecture only works on unweighted and undirected graphs.
>
> > Can the proposed method work well on directed and weighted graphs?
>
> **Directed graphs:** The inherent dilemma when it comes to directed graphs is that an arbitrarily long walk always exists on undirected graphs but not on directed graphs. As a temporary remedy, to get results on directed graphs in Tables 4 and 8, we have *symmetrized* the directed graph and annotated whether the walks are going *along* or *against* the inherent direction in the topological embedding $\omega_x$. Tables 4 and 8 show that this quick fix produces satisfactory performance---we plan to update the experiment details section to make this clearer. So RUM is already working well on (transformed) directed graphs.
>
> **Weighted graphs:** we excluded weighted graphs in **Section 4 Theory, Assumption 3** only for the simplicity and clarity of the _theoretical arguments_. In practice, running RUM on directed graphs is feasible with **biased random walk**, by multiplying the (unnormalized) random walk landing probability (Equation 4) with the edge weight $w_{ij}$:
>
> $P(v_j | (v_0, ..., v_{i-1})) \propto I [(v_i, v_j) \in E_G] * w_{ij} / D(v_i) $
>
> We plan to include this equation in our **Section 3: Architecture** to clarify that RUM can indeed run on weighted graphs.
>
> To sum up, RUM already works on transformed directed graphs and can be easily extended to weighted graphs. We hope that this has addressed your questions and you will consider increasing the score!

---

> > ### Author Response · Authors · 2024-08-08
> > **Directed and weighted graphs**
> >
> > Dear Reviewer `wXB7`,
> >
> > We believe that our rebuttal has sufficiently addressed your concern around the limitations when it comes to the weighted and directed graphs---both of which are possible with some slight modifications. Please let us know if you have any further questions!

---

> > > ### Comment · Reviewer_wXB7 · 2024-08-10
> > >
> > > Thanks authors for the response. I don't have other concerns, so I keep my score.

---

### Official Review · Reviewer_P1Jd · 2024-07-28

**Soundness:** 3
**Presentation:** 3
**Contribution:** 3
**Rating:** 7
**Confidence:** 4

**Summary:**

The authors of the paper proposed a non-convolution based approach to solve graph learning tasks using random walks and recurrent neural networks (RNNs), namely a random walk neural network with unifying memory (RUM). Random walks, together with the “anonymous experiment” function, allow to extract topological and semantic information from the graph, while an RNN and a feed-forward neural network take care of “unifying” the features collected from the walks. Research questions on the ability of the model to be more expressive than classical GNNs and to reduce oversmoothing and oversquashing have been answered positively both theoretically and experimentally. The experimental section, along with the appendix section, presents several experiments, ablation studies, training details, and theorem proofs. The code is available.

**Strengths:**

Originality: Novelty is fair/good. The proposed approach can be considered original by choosing to propose a model that does not use the graph convolution operator at all, which is the most widely used approach in the state of the art. In this way, it alleviates the problems associated with convolutions (e.g., oversmoothing, oversquashing) while maintaining or even improving performance. Random walks and RNNs (more specifically GRUs) are not new methodologies, but they are combined in an original way on graph learning.

Quality: The quality of the proposed paper is good. The claims and solutions are technically sound. They present theoretical and experimental proofs of their proposed approach.

Clarity: The clarity of the paper is good. It is well written and organized. The best results in the tables should be in bold or more visible.

Significance: The significance of the paper is medium-high. They proposed an effective method to avoid convolutions in GNNs that effectively outperforms and competes with SOTA methodologies.

**Weaknesses:**

Weakness: The paper lacks a discussion on the proposed method and Graph Transformer architectures. Although they are present in the experimental comparisons on the node classification task in Table 9, Graph Transformer is actually a well-known solution that does not rely on convolutions and can mitigate the over-squashing and oversmoothing problems [1,2]; the discussion should also considering the different computational costs of the two approaches.

Ref.
[1] "Attending to Graph Transformers", Luis Müller and Mikhail Galkin and Christopher Morris and Ladislav Rampášek, 2024, https://arxiv.org/abs/2302.04181
[2] "Do Transformers Really Perform Bad for Graph Representation?", Chengxuan Ying et. al., 2021, https://arxiv.org/abs/2106.05234

**Questions:**

1) At Table.5, showing Graph Regression comparisons on ESOL, FreeSolv and Lipophilicity datasets, it seems to me that for the MoleculeNet [1] benchmark you used the results presented in their papers about the XGBoost model (best performing among
conventional methods) with (0.99, 1.74, 0.799) RMSE scores, respectively to the three datasets, instead using them from the best performing graph-based methods which are (0.58,1.15) for ESOL and FreeSolv datasets by using the MPNN model and 0.655 for Lipophilicity dataset by using GC model, which are the best models in these datasets. This is in contrast on what you said in the caption of Table. 5, since those presented are not the best results and models of [1].

Ref.
[1] "MoleculeNet: a benchmark for molecular machine learning", Wu, Zhenqin and et. al., "Chem. Sci.",2018, doi:10.1039/C7SC02664A, url:http://dx.doi.org/10.1039/C7SC02664A

**Limitations:**

The authors have adequately discussed the potential negative societal impact and the current limitations of their work.

---

> ### Author Rebuttal · Authors · 2024-07-31
>
> Thank you, Reviewer `P1JD`, for your thorough and constructive review. We also thank you for recognizing our novelty in using simple building blocks to design novel methods. Based on your review, we plan to revise our manuscript to include a more comprehensive comparison with graph transformer architectures.
>
>
> ## Comparison with graph transformers.
> > Weakness: The paper lacks a discussion on the proposed method and Graph Transformer architectures.
>
> Many thanks for raising this issue. We plan to add the following paragraph to the **Related work** section to provide a conceptual discussion between graph transformers and RUM:
>
> ---
> **Graph transformers** \[1, 2\]---neural models that perform attention among all pairs of nodes and encode graph structure via positional encoding---are well-known solutions that are not *locally* convolutional. Their inductive biases determine that over-smoothing and over-squashing among local neighborhoods are, like RUM, also not prominent.
>
> Because of its all-to-all nature, the runtime complexity of graph transformers, just like that of almost all transformers, contains a quadratic term, w.r.t. the size of the system (number of nodes, edges, or subgraphs). This makes it prohibitively expensive and memory intensive on large social graphs (such as that used in Table 8 with millions of nodes). On smaller social graphs, we show in Table 9 that graph transformers are experimentally outperformed by RUM.
>
> ---
>
> We also thank you for pointing us to Ref 1. where there is some overlap in the benchmarking experiments, and we plan to include additional rows in Table 4 to compare with transformer-based models:
>
> |             | Texas         | Wisc.         | Cornell       |
> |-------------|---------------|---------------|---------------|
> | GCN         | 55.1 ± 4.2    | 51.8 ± 3.3    | 60.5 ± 4.8    |
> | GAT         | 52.2 ± 6.6    | 51.8 ± 3.1    | 60.5 ± 5.3    |
> | GCNII       | 77.6 ± 3.8    | 80.4 ± 3.4    | 77.9 ± 3.8    |
> | Geom-GCN    | 66.8 ± 2.7    | 64.5 ± 3.7    | 60.5 ± 3.7    |
> | PairNorm    | 60.3 ± 4.3    | 48.4 ± 6.1    | 58.9 ± 3.2    |
> | GPS| 75.4 ±1.5 | 78.0 ±2.9 | 65.4 ±5.7|
> |Transformer| 77.8 ±1.1 | 76.1 ±1.9 | 71.9 ±2.5 |
> |Graphomer| 76.8 ±1.8 | 77.7 ±2.0 | 68.4 ±1.7 |
> | RUM         | 80.0 ± 7.0    | 85.8 ± 4.1    | 71.1 ± 5.6    |
>
> The performance for GPS, Graphomer, and Transformer cited here is the best among various positional encodings. RUM outperforms all of them on Texas and Wisc. and is within the confidence interval of the best on Cornell.
>
> For the tasks in Ref. 2, we repeated the smaller MolHIV experiment (Table 3) using RUM and found that RUM performs similarly to Graphformer albeit with a significantly smaller parameter budget:
>
> | Method            | #param. | AUC (%)      |
> |-------------------|---------|--------------|
> | GCN-GraphNorm     | 526K    | 78.83 ± 1.00 |
> | PNA               | 326K    | 79.05 ± 1.32 |
> | PHC-GNN           | 111K    | 79.34 ± 1.16 |
> | DeeperGCN-FLAG    | 532K    | 79.42 ± 1.20 |
> | DGN               | 114K    | 79.70 ± 0.97 |
> | GIN-VN (fine-tune)| 3.3M    | 77.80 ± 1.82 |
> | Graphormer-FLAG   | 47.0M   | 80.51 ± 0.53 |
> | RUM | 87K | 80.01 ± 1.20 |
>
> ## Out-of-date SOTA reference
> > Questions: At Table.5, showing Graph Regression comparisons on ESOL, FreeSolv and Lipophilicity datasets, it seems to me that for the MoleculeNet [1] benchmark you used the results presented in their papers about the XGBoost model
>
> Many thanks, Reviewer `P1Jd`, for catching this. We were citing performance statistics from a perhaps outdated preprint. We will update these numbers with the new best scores in the correct reference. Though the margin now is smaller for RUM, this update does not change the main message of Table 5.

---

> > ### Author Response · Authors · 2024-08-08
> > **Comparison with graph transformers**
> >
> > Dear Reviewer `P1Jd`,
> >
> > We have included extensive conceptual discussion and additional experiments to incorporate your suggestion of comparing with graph transformer models. Please let us know if you have further questions, we will make sure to address them in the discussion period!

---

> > > ### Author Response · Authors · 2024-08-11
> > > **Conceptual and experimental comparison with graph transformers**
> > >
> > > Dear Reviewer `P1Jd`,
> > >
> > > As the discussion period is drawing to a close, we just wanted to bring your attention to the new conceptual and experimental comparison with graph transformers.
> > >
> > > Since this was the main part of the **weaknesses** section in your review, we would be grateful to know whether this concern has been addressed and if you would consider raising the score. If your concerns have not been addressed, please let us know and we will strive to provide more analysis, experiments, and discussion to provide more clarification!
> > >
> > > Thanks again for your valuable input!

---

> > > > ### Comment · Reviewer_P1Jd · 2024-08-11
> > > >
> > > > Thank you authors.  I don't have any concerns.  Based on your response and modification, I've increased the score.

---

> > > > > ### Author Response · Authors · 2024-08-11
> > > > > **Thank you!**
> > > > >
> > > > > Thank you, Reviewer `P1Jd` for your comment and for raising the score!

---

### Author Rebuttal · Authors · 2024-08-05

Thank you, all reviewers, for your constructive and detailed feedback, based on which we are further revising our manuscript for better clarity.

---
# Recap: Main contributions
With the common pitfalls of convolutional graph neural networks (GNN) identified, a new graph learning paradigm is designed, completely free of the convolution operator, coined *random walk with unifying memory* (RUM). With RUM, the topological (represented by _anonymous experiments_) and semantic features along random walks terminating at each node are merged by an RNN to form node representations.

We show that RUM has the following desired properties, with all experimentally validated and (1-3) also theoretically proven:
1. RUM is more expressive than ($k$-dimensional) Weisfeiler-Lehman ($k$-WL)-equivalent GNNs
2. RUM alleviates over-smoothing
3. RUM alleviates over-squashing
4. RUM has linear complexity w.r.t. the number of nodes and is faster than the simplest GNN on GPUs.
5. RUM is robust to random attacks on the graph structure.
6. RUM can be scaled to huge graphs with millions of nodes out-of-the-box and is naturally compatible with node mini-batching

In 8 tables totaling 21 benchmark experiments, we show that RUM achieves competitive performance across real-world graph- and node-level classification and regression tasks.

---
# New results in the rebuttal period: Graph transformer comparison, effect of the walk length, and $k$-WL comparison (see [the PDF](https://openreview.net/attachment?id=TStdjnCJG6&name=pdf))
## Graph Transformer comparison (`P1Jd`)
To compare RUM's performance with graph transformer models (`P1Jd`), Table 4 is expanded [in the separate PDF](https://openreview.net/attachment?id=TStdjnCJG6&name=pdf) to show that RUM outperforms graph transformer models on heterophilic datasets. A new table on `MolHIV` is also added to show that RUM achieves competitive performance on molecular graph regression tasks with a fraction of the parameter budget as Graphomer.

Conceptually, in the **Introduction & Related works** sections, RUM is currently being compared to convolutional GNNs, walk- and path-based GNNs, stochastic and constant-energy regularization methods, and graph rewiring methods. We thank Reviewer `P1JD` again for your valuable suggestion that RUM should be conceptually compared with graph transformers as they are well-known methods for solving over-smoothing and over-squashing that are also not locally convolutional. The discussion paragraph to be included in the updated manuscript can be found in this [separate rebuttal](https://openreview.net/forum?id=JDAQwysFOc&noteId=X8kKjbnpxZ).

---
## The effect of $L$, the length of the random walks, on molecular graph regression (`xhjS`)
An **Appendix Figure** is added to the PDF to show the impact of $L$ over the performance on a molecular graph regression task---ESOL. This also shows that for molecular graphs, because of the existence of long-term dependencies, the optimal $L$ is usually larger, compared to that of citation graphs (Figure 4). A recipe to narrow the hyperparameter search space of $L$ is provided [here](https://openreview.net/forum?id=JDAQwysFOc&noteId=4HQMSRau91).

---
## A more quantitative comparison with $k$-WL test (`xhjS`)
The **Corollary 4.1** is rewritten according to the suggestions from Reviewer `xhjS` to be quantitative linked with $k$-WL test:

**Corollary 4.1 (RUM is more expressive than k-WL-test.)** Up to the _Reconstruction Conjecture_, two graphs with $G_1 , G_2$ labeled as non-isomorphic by the $k$-dimensional Weisfeiler-Lehman ($k$-WL) isomorphism test, is the necessary, but not sufficient condition that the representations resulting from RUM with walk length $k$ are also different.

---


The rest of your suggestions are addressed as follows:

## Limitations: weighted and directed graphs. (`wXB7`)
With some slight modifications, RUM works on both *weighted* and *directed* graphs.

**Weighted graphs** are excluded in **Assumption 3** only for the clarity and simplicity of theoretical arguments. RUM can be easily extended to weighted graphs by sampling a **biased random walk** with edge weights $w_{ij}$ and keeping rest of the algorithm the same:
$P(v_j | (v_0, ..., v_{i-1})) \propto I [(v_i, v_j) \in E_G] * w_{ij} / D(v_i)$

**Directed graphs** are already included in the benchmarks (**Tables 4 and 8**, where RUM achieves competitive performance) with a slight modification---the edges are symmetrized with new, artificial edges annotated.

---
## Clarification on the non-diminishing Dirichlet energy (`jBUd`)
We plan to add more explanation around Lemma 6 to clarify that, higher Dirichlet energy doesn't mean better, and RUM does not *always* increase the Dirichlet energy but *is possible to maintain* a non-diminishing Dirichlet energy when non-contractive RNNs are prescribed as backbones, whereas for traditional convolutional GNNs, the Dirichlet energies are destined to diminish as the number of message-passing rounds increases (Figure 2).

---

Lastly, we apologize for the somewhat convoluting reading experience of this version of the manuscript (`jBUd`)---it has been drastically condensed to fit the page limit. As a result, many substantial results had to be moved to the **Appendix** and the **Introduction** section also doubles as a problem statement and a preliminary. We plan to straighten out many of the verbose sentences, reduce hyper-references and supplementary demonstration, and better highlight the best-performing models (`P1Jd`).

We hope that you will consider raising your score to see these valued suggestions of yours incorporated and representation improved in the next iteration of the manuscript.

Thank you again, all reviewers, for your time, effort, helpful feedback, and for helping us improve this manuscript together.

---

> ### Author Response · Authors · 2024-08-10
> **Correction: to remove the length condition in the new Corollary 4.1**
>
> After careful checking, we find that the condition of walk length $k$ has to be removed in the new Corollary 4.1 reworked during the rebuttal:
>
> **Corollary 4.1** (RUM is more expressive than k-WL-test.) Up to the Reconstruction Conjecture, two graphs with $G_1 , G_2$ labeled as non-isomorphic by the $k$-dimensional Weisfeiler-Lehman ($k$-WL) isomorphism test, is the necessary, but not sufficient condition that RUM can distinguish them.

---

### Author Response · Authors · 2024-08-09
**Please let us know if we have addressed your concerns**

Dear Reviewers and Area Chairs,

Please let us know if the rebuttal we provided has addressed your concerns. If you have more pending questions, we would be more than happy to carry out more analysis and experiments in the discussion period to address them!

Thank you again for your valuable suggestions!

---

> ### Comment · Area_Chair_bArP · 2024-08-10
>
> As the authors suggest (and due to the size of the rebuttal), I invite all reviewers to share their main concerns before the rebuttal deadline, to have sufficient time for discussion. I'd be especially interested in feedback from reviewer `jBUd`, who gave the most negative review as of now.
>
> Thanks, the AC

---

> ### Author Response · Authors · 2024-08-10
> **Thank you!**
>
> Thank you, Area Chair `bArP`, for inviting all reviewers to discuss the remaining concerns!

---

### Decision · Program_Chairs · 2024-09-25

**Decision:**

Accept (spotlight)

**Comment:**

The paper presents a non-convolutional graph layer, which extracts topological and semantic information from random walks over the graph and aggregates them with recurrent layers. They show the layer compares favorably to standard models, in terms of accuracy, complexity, and expressiveness.

The paper had a very favourable round of review, which further improved with the rebuttal - all reviewers recommend acceptance with no remaining concerns. The rebuttal has strengthened the paper, most notably with the inclusion of graph transformer baselines (reviewer `P1Jd`) and hyper-parameters selection (reviewer `xhjS`).

I think the paper has a good organization, the contribution is sufficiently novel, and the theoretical analysis is strong. Overall, I believe this is a very interesting paper for the GNN community at NeurIPS and I recommend its acceptance following some minor language editing (e.g., "reception" field or the first sentence in the abstract).